# Dynamic Threshold Global Performance-Guaranteed Formation Control for Wheeled Mobile Robots with Smooth Extended State Observer

1st Minjing Wang
*School of Information and Communication Engineering*
*Hainan University*
Haikou, China
mjwang@hainanu.edu.cn

2nd Di Wu
*School of Information and Communication Engineering*
*Hainan University*
Haikou, China
hainuwudi@hainanu.edu.cn

3rd Yibo Zhang
*Department of Automation*
*Shanghai Jiao Tong University*
Shanghai, China
zhang297@sjtu.edu.cn

4th Wenlong Feng
*School of Information and Communication Engineering*
*Hainan University*
Haikou, China
fwlfwl@163.com

*Abstract*—In this paper, a dynamic threshold global performance-guaranteed formation control method is proposed for wheeled mobile robots (WMRs). Unlike existing prescribed performance formation control methods that are constrained by initial values, we design a dynamic threshold global performance-guaranteed (DTGPG) function to address the initial value constraints while being able to secondary adjust the steady state performance boundaries. Moreover, we design a smooth extended state observer (SESO) based on a sigmoid-like function to mitigate the chattering problem of the existing event-triggered ESO. Then a DTGPG-based guidance law and a SESO-based control law are designed to implement the formation control. The proof shows that the total closed-loop system is input-to-state stable (ISS). Through simulation, the benefits and validity of the proposed control methodology are confirmed.

*Index Terms*—WMRs, dynamic threshold global performance-guaranteed function, formation control, SESO

## I. INTRODUCTION

Multi-wheeled mobile robots (WMRs) formation control with extremely high demands on transient and steady state performance. In the transient phase, small overshoots and fast convergence can avoid collisions between WMRs. In the steady state phase, high accuracy tracking performance can significantly improve the overall coordination and task execution efficiency. Therefore, it is crucial to prescribe the performance of the multi-WMRs system. In [1], a collision avoidance prescribed performance control (PPC) method is proposed for WMR formations, which guarantees the performance of the multi-WMR system by adding communication limits and collision limits to the prescribed performance

This work is partly distributed under the "South China Sea Rising Star" Education Platform Foundation of Hainan Province (JYNHXX2023-17G), the Natural Science Foundation of Hainan Province (624MS036). (Corresponding author: Di Wu)

function. In [2], a fixed-time performance-guaranteed formation control problem for multi-WMRs is investigated, which achieves fixed-time convergence by introducing a segmented time-varying function into the performance function. In [3], a field-of-view constrained performance-guaranteed formation control method is proposed for multi-WMRs, which designs a guaranteed performance function that considers leader and follower distance maintenance to avoid collisions. Although the above work [1]–[3] can effectively improve the performance of multi-WMRs, there are still two points that need to be improved: 1. They are all subject to initial conditions, which will increase the human intervention in practical applications, i.e., calculating the starting position of the WMRs in advance. 2. The standard PPC cannot perform a secondary adjustment of the performance boundaries after reaching the steady state.

On the other hand, when performing tasks in complex environments, frozen and uneven road surfaces are usually encountered. These disturbances may affect the stability of WMR formations. Therefore, how to quickly and accurately estimate the external disturbances is also crucial. In [4], a nonlinear extended state observer (ESO) is proposed to estimate the external disturbance, which recovers the velocity and estimates the external disturbance through position and heading errors. Then to improve the estimation rate, a finite time ESO is designed. In [5], an event-triggered ESO is designed to adjust the allocation of resources. Note that event-triggered ESO [5] can save resources when estimating disturbances, but will inevitably have chattering problems.

Inspired by the aforementioned observations, we propose a dynamic threshold global performance-guaranteed (DTGPG) formation control method for WMRs with a smooth extended state observer (SESO). The key contributions of this work are: Unlike the standard PPC methods described in [6] and the

TABLE I
SYMBOL DEFINITION

| Symbol | Definition |
|---|---|
| $\mathbb{R}^n$ | $n$-dimensional Euclidean Space |
| $\mathbb{R}^+$ | Positive real space |
| $\|\cdot\|$ | Euclidean norm |
| $\mathrm{diag}\{\cdots\}$ | Block-diagonal matrix |
| $\lambda_{\max}(\cdot)$ | Maximum eigenvalue of a matrix |
| $\lambda_{\min}(\cdot)$ | Minimum eigenvalue of a matrix |
| $\mathrm{sgn}(\cdot)$ | Sign function |
| $\exp(\cdot)$ | Exponential function |
| $\mathrm{col}(\cdot)$ | Column vector |

TPP methods in [7]–[9], this paper proposes DTGPG capable of solving the initial value constraints problem and secondary adjustment of the steady state performance bounds. In contrast to event-triggered ESO [5], we design the SESO to mitigate chattering by introducing a sigmoid-like function to smooth the estimation error. The total closed-loop system is proved to be input-to-state stable (ISS). Some of the symbols in this paper are defined in Table I.

## II. PRELIMINARIES AND PROBLEM STATEMENT

### A. Graph Theory

To describe the communication among the virtual leader and WMRs, a directed graph is described as $\mathcal{G} = \{\mathcal{V}, \mathcal{M}\}$. $\mathcal{V} = \{n_1, \ldots, n_M\}$ and $\mathcal{M} = \{(n_i, n_j) \in \mathcal{V} \times \mathcal{V}\}$ represent a vertex set and an edge set, respectively. An adjacency matrix associated with $\mathcal{G}$ is defined as $\mathcal{A} = [a_{ij}] \in \mathbb{R}^{M \times M}$. Correspondingly, a degree matrix connected with $\mathcal{G}$ is characterized as $\mathcal{D} = \mathrm{diag}\{d_i\} \in \mathbb{R}^{M \times M}$ with $d_i = \sum_{j=1}^{M} a_{ij}$. Additionally, a Laplacian matrix associated with $\mathcal{G}$ is defined as $\mathcal{L} = \mathcal{D} - \mathcal{A}$. Note that here $i = 1, ..., M, j = 1, ..., M$.

### B. Problem Statement

Suppose that there exist $N$ followers, labeled as agents $n_1$ to $n_N$, and $M - N$ leaders, labeled as agents $n_{N+1}$ to $n_M$, under a communication topology graph. A group of followers consisting of $N$ wheeled mobile robots is modelled as follows

$$
\begin{cases}
\dot{\boldsymbol{\eta}}_i = \boldsymbol{R}_i \boldsymbol{\nu}_i \\
\dot{\boldsymbol{\nu}}_i = r_i \boldsymbol{J}_i^+ \boldsymbol{M}_i^{-1} \boldsymbol{\tau}_i + r_i \boldsymbol{J}_i^+ \boldsymbol{M}_i^{-1} \boldsymbol{\mathcal{T}}_i \\
\quad - D_{i\theta} r_i^2 \boldsymbol{J}_i^+ \boldsymbol{M}_i^{-1} \boldsymbol{J}_i \boldsymbol{R}_i^{-1} \dot{\boldsymbol{\eta}}_i - \boldsymbol{J}_i^+ \boldsymbol{M}_i^{-1} \boldsymbol{\mathcal{F}}_i r_i^2
\end{cases} \quad (1)
$$

where $i=1, ..., N$. $\boldsymbol{\eta}_i = [x_i, y_i, \psi_i]^T \in \mathbb{R}^3$ denotes the position and yaw angle. $\boldsymbol{\nu}_i = [u_i, v_i, w_i]^T \in \mathbb{R}^3$ denotes the velocity vector. $\boldsymbol{\tau}_i = [\tau_{i1}, \tau_{i2}, \tau_{i3}, \tau_{i4}]^T \in \mathbb{R}^4$ denotes the control input. $\boldsymbol{\mathcal{T}}_i = [\mathcal{T}_{i1}, \mathcal{T}_{i2}, \mathcal{T}_{i3}, \mathcal{T}_{i4}]^T \in \mathbb{R}^4$ denotes the external disturbance. The kinetic parameters and matrices of this WMR can be found in [10]. $\boldsymbol{J}_i \in \mathbb{R}^{4 \times 3}$ and $\boldsymbol{J}_i^+ \in \mathbb{R}^{3 \times 4}$ satisfy the relationship $\boldsymbol{J}_i^+ \boldsymbol{J}_i = \boldsymbol{I}_3$.

*Assumption 1:* The graph $\mathcal{G}$ contains a spanning tree with the virtual leader as the root node.

### C. Dynamic Threshold Global Performance-Guaranteed and Barrier Function

We define the distributed error as follows

$$
\boldsymbol{E}_i = \sum_{j=1}^{N} a_{ij}(\boldsymbol{\eta}_i - \boldsymbol{\eta}_j) + \sum_{j=N+1}^{M} a_{ij}(\boldsymbol{\eta}_i - \boldsymbol{\eta}_{jr}) \quad (2)
$$

where $\boldsymbol{\eta}_{jr} = [\eta_{jx}, \eta_{jy}, \eta_{j\psi}]^T \in \mathbb{R}^3$ represents the trajectory of the virtual leader. The coefficient $a_{ij}$ is defined in [11]. To ensure that the developed control is free from the influence of initial conditions and can dynamically adjust prescribed thresholds, the error is constrained within the following prescribed regions

$$
\mathcal{I}_{ik}(-\mathcal{W}_{ik}) \leq E_{ik} \leq \mathcal{I}_{ik}(\mathcal{W}_{ik}), \quad k = x, y, \psi \quad (3)
$$

where $\mathcal{I}_{ik}(\mathcal{W}_{ik})$ is a dynamic threshold global performance-guaranteed (DTGPG) function similar to the [12], and is defined as follows

$$
\mathcal{I}_{ik}(\mathcal{W}_{ik}) = \frac{\sqrt{l_{ik}} \mathcal{W}_{ik}}{\sqrt{1 - \mathcal{W}_{ik}^2}} \quad (4)
$$

with $\mathcal{W}_{ik} = 1/\mathcal{P}_{ik}$. $\mathcal{P}_{ik}$ is a dynamic threshold finite-time prescribed function similar to the [13]

$$
\mathcal{P}_{ik}(t) = \begin{cases}
(1 - \Theta_{ik,\infty}) \exp(-\varrho_{ik} \frac{T_{ik,a} t}{T_{ik,a} - t}) + \Theta_{ik,\infty}, 0 \leq t < T_{ik,a} \\
\Theta_{ik,\infty}(1 - \frac{\omega_{ik}}{2} + \frac{\omega_{ik}}{2} \cos(\frac{\pi}{c_{ik}}(t - T_{ik,a}))), T_{ik,a} \leq t < T_{ik,b} \\
\Theta_{ik,\infty}(1 - \omega_{ik}), \qquad\qquad t \geq T_{ik,b}
\end{cases} \quad (5)
$$

where $l_{ik}$ and $\omega_{ik}$ are positive constants. $\Theta_{ik,\infty} = \lim_{t \to \infty} \Theta_{ik}(t)$ is the steady-state value. $\varrho_{ik} > 0$ represents the convergence rate. $T_{ik,a}$ is the settling time to reach steady state. $c_{ik} = T_{ik,b} - T_{ik,a}$ is the duration of the dynamic adjustment.

Then, we employ the following barrier function to implement the error constraint in (3)

$$
\mathcal{Z}_{ik} = \frac{\mathcal{J}_{ik}}{1 - \mathcal{J}_{ik}^2} \quad (6)
$$

where $\mathcal{J}_{ik} = \mathcal{P}_{ik} \mathcal{H}_{ik}$ with $\mathcal{H}_{ik} = E_{ik}/\sqrt{E_{ik}^2 + l_{ik}}$. The properties of the barrier function are described in [12].

## III. CONTROLLER DESIGN AND ANALYSIS

### A. Smooth Extended State Observer

To facilitate the subsequent strategy design, define $\boldsymbol{\Lambda}_i = r_i \boldsymbol{J}_i^+ \boldsymbol{M}_i^{-1} \boldsymbol{\mathcal{T}}_i - D_{i\theta} r_i^2 \boldsymbol{J}_i^+ \boldsymbol{M}_i^{-1} \boldsymbol{J}_i \boldsymbol{R}_i^{-1} \dot{\boldsymbol{\eta}}_i - \boldsymbol{J}_i^+ \boldsymbol{M}_i^{-1} \boldsymbol{\mathcal{F}}_i r_i^2$ to denote internal uncertainty and external disturbances suffered by the $i$th WMR. (1) can be reformulated as

$$
\begin{cases}
\dot{\boldsymbol{\eta}}_i = \boldsymbol{R}_i \boldsymbol{\nu}_i \\
\dot{\boldsymbol{\nu}}_i = r_i \boldsymbol{J}_i^+ \boldsymbol{M}_i^{-1} \boldsymbol{\tau}_i + \boldsymbol{\Lambda}_i.
\end{cases} \quad (7)
$$

*Assumption 2:* For the multi-WMR system, the unknown total disturbance $\boldsymbol{\Lambda}_i$ is smooth and continuous.

Then, we regard the total disturbances $\boldsymbol{\Lambda}_i$ as an extended state, and to avoid unnecessary waste of resources when approximating the disturbances, an ESO based on event-triggered mechanism is designed as [5]

$$
\begin{cases}
\tilde{\boldsymbol{\nu}}_i^s = \hat{\boldsymbol{\nu}}_i - \boldsymbol{\nu}_i^{\star} \\
\dot{\hat{\boldsymbol{\nu}}}_i = -\boldsymbol{\varepsilon}_{i1}\tilde{\boldsymbol{\nu}}_i^s + \hat{\boldsymbol{\Lambda}}_i + r_i \boldsymbol{J}_i^+ \boldsymbol{M}_i^{-1}\boldsymbol{\tau}_i \\
\dot{\hat{\boldsymbol{\Lambda}}}_i = -\boldsymbol{\varepsilon}_{i2}\tilde{\boldsymbol{\nu}}_i^s
\end{cases}
\tag{8}
$$

where $\boldsymbol{\varepsilon}_{i1}$ and $\boldsymbol{\varepsilon}_{i2}\in\mathbb{R}^{3\times3}$ denote positive diagonal matrices. The variables $\hat{\boldsymbol{\nu}}_i=[\hat{u}_i,\hat{v}_i,\hat{w}_i]^T\in\mathbb{R}^3$ and $\hat{\boldsymbol{\Lambda}}_i=[\hat{\Lambda}_{iu},\hat{\Lambda}_{iv},\hat{\Lambda}_{iw}]^T\in\mathbb{R}^3$ denote the estimates of $\boldsymbol{\nu}_i$ and $\boldsymbol{\Lambda}_i$, respectively. $\boldsymbol{\nu}_i^{\star}\in\mathbb{R}^3$ represents the aperiodic sampling of $\boldsymbol{\nu}_i$. The event-triggered mechanism is defined as

$$
\begin{cases}
\boldsymbol{\nu}_i^{\star}(t)=\boldsymbol{\nu}_i(t_\varpi^{\nu_i}),\forall t\in[t_\varpi^{\nu_i},t_{\varpi+1}^{\nu_i}),\tilde{\boldsymbol{\nu}}_{is}(t)=\boldsymbol{\nu}_i^{\star}(t)-\boldsymbol{\nu}_i(t) \\
t_{\varpi+1}^{\nu_i} = \inf\{t\in\mathbb{R}\mid\|\tilde{\boldsymbol{\nu}}_{is}(t)\|\geq\mathcal{X}_i\}
\end{cases}
\tag{9}
$$

where $\mathcal{X}_i\in\mathbb{R}^+$ denotes the event triggering threshold, and $\tilde{\boldsymbol{\nu}}_{is}(t)$ denotes the aperiodic sampling error. When $\|\tilde{\boldsymbol{\nu}}_{is}(t)\|\geq\mathcal{X}_i$, update $\boldsymbol{\nu}_i^{\star}(t)$; otherwise, maintain the last updated value.

*Remark 1:* In addition to using ESO to estimate the external disturbances, the neural network [14] and the neural predictor [15] also achieve the same objective.

Existing ESO based on event-triggered mechanism [5] suffers from unavoidable chattering when approximating the disturbances. To solve the chattering problem, we design the SESO as follows

$$
\begin{cases}
\dot{\hat{\boldsymbol{\nu}}}_i = -\boldsymbol{\varepsilon}_{i1}\tilde{\boldsymbol{\nu}}_i^s + \hat{\boldsymbol{\Lambda}}_i + r_i \boldsymbol{J}_i^+ \boldsymbol{M}_i^{-1}\boldsymbol{\tau}_i \\
\dot{\hat{\boldsymbol{\Lambda}}}_i = -\boldsymbol{\varepsilon}_{i2}\boldsymbol{\mathcal{B}}(\tilde{\boldsymbol{\nu}}_i^s)
\end{cases}
\tag{10}
$$

where $\boldsymbol{\mathcal{B}}(\tilde{\boldsymbol{\nu}}_i^s) = \mathrm{col}(\mathcal{B}(\tilde{\nu}_{i\Xi}^s))$, $\Xi = u,v,w \in\mathbb{R}^3$ is the sigmoid-like function vector, defined as follows

$$
\mathcal{B}(\tilde{\nu}_{i\Xi}^s) = \begin{cases}
\dfrac{1-\exp(-|\tilde{\nu}_{i\Xi}^s|)}{1+\exp(-|\tilde{\nu}_{i\Xi}^s|)}\dfrac{\tilde{\nu}_{i\Xi}^s}{|\tilde{\nu}_{i\Xi}^s|}, & \tilde{\nu}_{i\Xi}^s\neq0 \\
\tilde{\nu}_{i\Xi}^s, & \tilde{\nu}_{i\Xi}^s=0.
\end{cases}
\tag{11}
$$

Next, to facilitate the stability analysis of the SESO, define a positive vector $\boldsymbol{\mathcal{V}}_i = \mathrm{diag}\{\mathcal{V}_{i\Xi}\}\in\mathbb{R}^{3\times3}$ with

$$
\mathcal{V}_{i\Xi} = \begin{cases}
\dfrac{1-\exp(-|\tilde{\nu}_{i\Xi}^s|)}{1+\exp(-|\tilde{\nu}_{i\Xi}^s|)}\dfrac{1}{|\tilde{\nu}_{i\Xi}^s|}, & \tilde{\nu}_{i\Xi}^s\neq0 \\
1, & \tilde{\nu}_{i\Xi}^s=0.
\end{cases}
\tag{12}
$$

The (10) can be rewritten as

$$
\begin{cases}
\dot{\hat{\boldsymbol{\nu}}}_i = -\boldsymbol{\varepsilon}_{i1}\tilde{\boldsymbol{\nu}}_i + \boldsymbol{\varepsilon}_{i1}\tilde{\boldsymbol{\nu}}_{is} + \hat{\boldsymbol{\Lambda}}_i + r_i \boldsymbol{J}_i^+ \boldsymbol{M}_i^{-1}\boldsymbol{\tau}_i \\
\dot{\hat{\boldsymbol{\Lambda}}}_i = -\boldsymbol{\varepsilon}_{i2}\boldsymbol{\mathcal{V}}_i\tilde{\boldsymbol{\nu}}_i + \boldsymbol{\varepsilon}_{i2}\boldsymbol{\mathcal{V}}_i\tilde{\boldsymbol{\nu}}_{is}
\end{cases}
\tag{13}
$$

where $\tilde{\boldsymbol{\nu}}_i = \hat{\boldsymbol{\nu}}_i - \boldsymbol{\nu}_i$, $\tilde{\boldsymbol{\Lambda}}_i = \hat{\boldsymbol{\Lambda}}_i - \boldsymbol{\Lambda}_i$. Define $\boldsymbol{\mathcal{N}}_{i1} = [\tilde{\boldsymbol{\nu}}_i,\tilde{\boldsymbol{\Lambda}}_i]^T\in\mathbb{R}^6$, one has

$$
\dot{\boldsymbol{\mathcal{N}}}_{i1} = \boldsymbol{A}_{i1}\boldsymbol{\mathcal{N}}_{i1} + \boldsymbol{B}_{i1}\tilde{\boldsymbol{\nu}}_{is} + \boldsymbol{C}_{i1}\dot{\boldsymbol{\Lambda}}_i
\tag{14}
$$

where

$$
\begin{cases}
\boldsymbol{A}_{i1} = \begin{bmatrix} -\boldsymbol{\varepsilon}_{i1}\boldsymbol{I}_3 & \boldsymbol{I}_3 \\ -\boldsymbol{\varepsilon}_{i2}\boldsymbol{\mathcal{V}}_i & \boldsymbol{O}_3 \end{bmatrix} \boldsymbol{B}_{i1} = \begin{bmatrix} \boldsymbol{\varepsilon}_{i1}\boldsymbol{I}_3 \\ \boldsymbol{\varepsilon}_{i2}\boldsymbol{\mathcal{V}}_i \end{bmatrix} \boldsymbol{C}_{i1} = \begin{bmatrix} \boldsymbol{O}_3 \\ \boldsymbol{I}_3 \end{bmatrix}.
\end{cases}
$$

Note that the matrix $\boldsymbol{A}_{i1}$ is a Hurwitz matrix. There exists a positive-definite matrix $\boldsymbol{P}_{i1}$ satisfying the following inequality

$$
\boldsymbol{A}_{i1}^T\boldsymbol{P}_{i1} + \boldsymbol{P}_{i1}\boldsymbol{A}_{i1} \leq -\jmath_{i1}\boldsymbol{I}_6.
\tag{15}
$$

*Lemma 1:* The system (14) is ISS.

*Proof:* Consider a Lyapunov function candidate as follows

$$
V_1 = \frac{1}{2}\sum_{i=1}^{N}\boldsymbol{\mathcal{N}}_{i1}^T\boldsymbol{P}_{i1}\boldsymbol{\mathcal{N}}_{i1}.
\tag{16}
$$

The time derivative $V_1$ based on (14) and (15) satisfies

$$
\dot{V}_1 \leq -\frac{\jmath_1}{2}\|\boldsymbol{\mathcal{N}}_1\|^2 + \|\boldsymbol{\mathcal{N}}_1\|\|\boldsymbol{P}_1\boldsymbol{B}_1\|\|\tilde{\boldsymbol{\nu}}_s\| \\
+ \|\boldsymbol{\mathcal{N}}_1\|\|\boldsymbol{P}_1\boldsymbol{C}_1\|\|\dot{\boldsymbol{\Lambda}}\|
\tag{17}
$$

where $\jmath_1=\min_{i=1,...,N}(\jmath_{i1})$, $\boldsymbol{\mathcal{N}}_1=[\boldsymbol{\mathcal{N}}_{11}^T,...,\boldsymbol{\mathcal{N}}_{N1}^T]^T\in\mathbb{R}^{6N}$, $\tilde{\boldsymbol{\nu}}_s=[\tilde{\boldsymbol{\nu}}_{1s}^T,...,\tilde{\boldsymbol{\nu}}_{Ns}^T]^T\in\mathbb{R}^{3N}$, $\dot{\boldsymbol{\Lambda}} = [\dot{\boldsymbol{\Lambda}}_1^T,...,\dot{\boldsymbol{\Lambda}}_N^T]^T \in\mathbb{R}^{3N}$, $\boldsymbol{P}_1=\mathrm{diag}\{\boldsymbol{P}_{11},...,\boldsymbol{P}_{N1}\}\in\mathbb{R}^{6N\times6N}$, $\boldsymbol{B}_1=\mathrm{diag}\{\boldsymbol{B}_{11},...,\boldsymbol{B}_{N1}\}\in\mathbb{R}^{6N\times3N}$, and $\boldsymbol{C}_1 = \mathrm{diag}\{\boldsymbol{C}_{11},...,\boldsymbol{C}_{N1}\}\in\mathbb{R}^{6N\times3N}$. Since $\|\boldsymbol{\mathcal{N}}_1\| \geq 2(\|\boldsymbol{P}_1\boldsymbol{B}_1\|\|\tilde{\boldsymbol{\nu}}_s\| + \|\boldsymbol{P}_1\boldsymbol{C}_1\|\|\dot{\boldsymbol{\Lambda}}\|)/\jmath_1\sigma_1$, one has $\dot{V}_1 \leq -\jmath_1(1-\sigma_1)\|\boldsymbol{\mathcal{N}}_1\|^2/2$, where $0 < \sigma_1 < 1$. It follows that the subsystem (14) is ISS. There exists a $\mathcal{KL}$ function $\mathcal{Y}_1(\cdot)$ and $\mathcal{K}_\infty$ function $\mathcal{C}^{\tilde{\boldsymbol{\nu}}_s}(\cdot)$ and $\mathcal{C}^{\dot{\boldsymbol{\Lambda}}}(\cdot)$ satisfying $\|\boldsymbol{\mathcal{N}}_1(t)\| \leq \mathcal{Y}_1(\|\boldsymbol{\mathcal{N}}_1(0)\|,t) + \mathcal{C}^{\tilde{\boldsymbol{\nu}}_s}(\|\tilde{\boldsymbol{\nu}}_s\|) + \mathcal{C}^{\dot{\boldsymbol{\Lambda}}}(\|\dot{\boldsymbol{\Lambda}}\|)$, where $\mathcal{C}^{\tilde{\boldsymbol{\nu}}_s}(s) = ((2s\|\boldsymbol{P}_1\boldsymbol{B}_1\|\sqrt{\lambda_{\max}(\boldsymbol{P}_1)})/(\jmath_1\sigma_1\sqrt{\lambda_{\min}(\boldsymbol{P}_1)}))$ and $\mathcal{C}^{\dot{\boldsymbol{\Lambda}}}(s) = ((2s\|\boldsymbol{P}_1\boldsymbol{C}_1\|\sqrt{\lambda_{\max}(\boldsymbol{P}_1)})/(\jmath_1\sigma_1\sqrt{\lambda_{\min}(\boldsymbol{P}_1)}))$.

### B. Design of Guidance Law and Control Law

In this section, we design the DTGPG-based guidance law and the SESO-based control law. First, we design the guidance law. The time derivative of (6) is represented by

$$
\dot{\mathcal{Z}}_{ik} = \mu_{ik}\mathcal{P}_{ik}\rho_{ik}\dot{E}_{ik} + \mu_{ik}\dot{\mathcal{P}}_{ik}\mathcal{H}_{ik}
\tag{18}
$$

where $\mu_{ik} = (1 + \mathcal{J}_{ik}^2)/(1 - \mathcal{J}_{ik}^2)^2$ and $\rho_{ik} = l_{ik}/(\sqrt{E_{ik}^2 + l_{ik}}(E_{ik}^2 + l_{ik}))$.

Next, to simplify the design of the controller, we rewrite (18) in a vector form

$$
\dot{\boldsymbol{\mathcal{Z}}}_i = \boldsymbol{\mu}_{i1}\dot{\boldsymbol{E}}_i + \boldsymbol{\mu}_{i2}
\tag{19}
$$

where $\boldsymbol{\mathcal{Z}}_i = [\mathcal{Z}_{ix},\mathcal{Z}_{iy},\mathcal{Z}_{i\psi}]^T\in\mathbb{R}^3$, $\boldsymbol{E}_i = [E_{ix},E_{iy},E_{i\psi}]^T\in\mathbb{R}^3$, $\boldsymbol{\mu}_{i1} = \mathrm{diag}\{\mu_{ix}\mathcal{P}_{ix}\rho_{ix},\mu_{iy}\mathcal{P}_{iy}\rho_{iy},\mu_{i\psi}\mathcal{P}_{i\psi}\rho_{i\psi}\}\in\mathbb{R}^{3\times3}$, and $\boldsymbol{\mu}_{i2}=\mathrm{diag}\{\mu_{ix}\dot{\mathcal{P}}_{ix}\mathcal{H}_{ix},\mu_{iy}\dot{\mathcal{P}}_{iy}\mathcal{H}_{iy},\mu_{i\psi}\dot{\mathcal{P}}_{i\psi}\mathcal{H}_{i\psi}\}\in\mathbb{R}^{3\times3}$.

Take the time derivative of (2) based on (1) satisfies

$$
\dot{\boldsymbol{E}}_i = \iota_i\boldsymbol{R}_i\boldsymbol{\nu}_i - \sum_{j=1}^{N}a_{ij}\boldsymbol{R}_j\boldsymbol{\nu}_j - \sum_{j=N+1}^{M}a_{ij}\dot{\boldsymbol{\eta}}_{jr}
\tag{20}
$$

where $\iota_i = \sum_{j=1}^{N}a_{ij} + \sum_{j=N+1}^{M}a_{ij}$. Substituting (20) into (19) results in

$$
\dot{\boldsymbol{\mathcal{Z}}}_i = \boldsymbol{\mu}_{i1}(\iota_i\boldsymbol{R}_i\boldsymbol{\nu}_i - \sum_{j=1}^{N}a_{ij}\boldsymbol{R}_j\boldsymbol{\nu}_j - \sum_{j=N+1}^{M}a_{ij}\dot{\boldsymbol{\eta}}_{jr}) + \boldsymbol{\mu}_{i2}.
\tag{21}
$$

From (21), the DTGPG-based guidance law is chosen as

$$
\boldsymbol{\alpha}_i = \frac{1}{\iota_i\boldsymbol{R}_i}(\sum_{j=1}^{N}a_{ij}\boldsymbol{R}_j\hat{\boldsymbol{\nu}}_j + \sum_{j=N+1}^{M}a_{ij}\dot{\boldsymbol{\eta}}_{jr} - \frac{1}{\boldsymbol{\mu}_{i1}}(\boldsymbol{\kappa}_{i1}\boldsymbol{\mathcal{Z}}_i + \boldsymbol{\mu}_{i2})).
\tag{22}
$$

We substitute (22) into (21), and it follows that

$$\dot{\boldsymbol{\mathcal{Z}}}_i = \boldsymbol{\mu}_{i1} \sum_{j=1}^{N} a_{ij} \boldsymbol{R}_j \tilde{\boldsymbol{\nu}}_j - \boldsymbol{\kappa}_{i1} \boldsymbol{\mathcal{Z}}_i \qquad (23)$$

with $\boldsymbol{\kappa}_{i1} \in \mathbb{R}^{3 \times 3}$ being a positive diagonal matrix.

Differing from the first-order low-pass filtering method in the traditional DSC, a second-order linear tracking differentiator (LTD) with respect to $\boldsymbol{\alpha}_i$ is introduced

$$\begin{cases} \dot{\boldsymbol{\alpha}}_{if} = \boldsymbol{\alpha}_{if}^* \\ \dot{\boldsymbol{\alpha}}_{if}^* = -\gamma_i^2 \big( (\boldsymbol{\alpha}_{if} - \boldsymbol{\alpha}_i) + 2(\boldsymbol{\alpha}_{if}^*/\gamma_i) \big) \end{cases} \qquad (24)$$

where $\boldsymbol{\alpha}_{if}^* \in \mathbb{R}^3$ is the filtered value of $\dot{\boldsymbol{\alpha}}_i$, and $\gamma_i \in \mathbb{R}^+$.

Second, we design the control law. Defining a velocity error $\boldsymbol{\mathcal{Z}}_{ie} = \boldsymbol{\nu}_i - \boldsymbol{\alpha}_i \in \mathbb{R}^3$, $\dot{\boldsymbol{\mathcal{Z}}}_{ie}$ along (7) satisfies

$$\dot{\boldsymbol{\mathcal{Z}}}_{ie} = r_i \boldsymbol{J}_i^+ \boldsymbol{M}_i^{-1} \boldsymbol{\tau}_i + \boldsymbol{\Lambda}_i - \dot{\boldsymbol{\alpha}}_i. \qquad (25)$$

Then, we designed the SESO-based control law to stabilize (25)

$$\boldsymbol{\tau}_i = \frac{\boldsymbol{M}_i \boldsymbol{J}_i}{r_i} \left( \boldsymbol{\alpha}_{if}^* - \hat{\boldsymbol{\Lambda}}_i - \boldsymbol{\kappa}_{i2} \boldsymbol{\mathcal{Z}}_{ie} \right) \qquad (26)$$

with $\boldsymbol{\kappa}_{i2} \in \mathbb{R}^{3 \times 3}$ being a positive diagonal matrix.

The dynamics of $\boldsymbol{\mathcal{Z}}_{ie}$ is further obtained by substituting (26) into (25)

$$\dot{\boldsymbol{\mathcal{Z}}}_{ie} = \tilde{\boldsymbol{\alpha}}_i^* - \tilde{\boldsymbol{\Lambda}}_i - \boldsymbol{\kappa}_{i2} \boldsymbol{\mathcal{Z}}_{ie} \qquad (27)$$

where $\tilde{\boldsymbol{\alpha}}_i^* = \boldsymbol{\alpha}_{if}^* - \dot{\boldsymbol{\alpha}}_i$.

From (23) and (27), we can obtain the following subsystems

$$\begin{cases} \dot{\boldsymbol{\mathcal{Z}}}_i = \boldsymbol{\mu}_{i1} \sum_{j=1}^{N} a_{ij} \boldsymbol{R}_j \tilde{\boldsymbol{\nu}}_j - \boldsymbol{\kappa}_{i1} \boldsymbol{\mathcal{Z}}_i \\ \dot{\boldsymbol{\mathcal{Z}}}_{ie} = \tilde{\boldsymbol{\alpha}}_i^* - \tilde{\boldsymbol{\Lambda}}_i - \boldsymbol{\kappa}_{i2} \boldsymbol{\mathcal{Z}}_{ie}. \end{cases} \qquad (28)$$

*Lemma 2:* The system (28) is ISS.

*Proof:* Consider a Lyapunov function candidate as $V_2 = (1/2) \sum_{i=1}^{N} (\boldsymbol{\mathcal{Z}}_i^T \boldsymbol{\mathcal{Z}}_i + \boldsymbol{\mathcal{Z}}_{ie}^T \boldsymbol{\mathcal{Z}}_{ie})$. The time derivative of $V_2$ based on (28) satisfies

$$\begin{aligned} \dot{V}_2 \leq &-n_1 \|\boldsymbol{\mathcal{Z}}\|^2 - n_2 \|\boldsymbol{\mathcal{Z}}_e\|^2 + n_3 n^* \|\boldsymbol{\mathcal{Z}}\| \|\tilde{\boldsymbol{\nu}}\| \\ &+ \|\boldsymbol{\mathcal{Z}}_e\| \|\tilde{\boldsymbol{\alpha}}^*\| + \|\boldsymbol{\mathcal{Z}}_e\| \|\tilde{\boldsymbol{\Lambda}}\| \end{aligned} \qquad (29)$$

where $n_1 = \lambda_{\min}(\boldsymbol{\kappa}_1)$ with $\boldsymbol{\kappa}_1 = \mathrm{diag}\{\boldsymbol{\kappa}_{11}, ..., \boldsymbol{\kappa}_{N1}\} \in \mathbb{R}^{3N \times 3N}$. $n_2 = \lambda_{\min}(\boldsymbol{\kappa}_2)$ with $\boldsymbol{\kappa}_2 = \mathrm{diag}\{\boldsymbol{\kappa}_{12}, ..., \boldsymbol{\kappa}_{N2}\} \in \mathbb{R}^{3N \times 3N}$. $n_3 = \max_{i=1,...,N}(\lambda_{\max}(\boldsymbol{\mu}_{i1}))$. $n^* = \max_{i=1,...,N}(n_i^*)$ with $n_i^* = \sum_{j=1}^{N} a_{ji}$. $\boldsymbol{\mathcal{Z}} = [\boldsymbol{\mathcal{Z}}_1^T, ..., \boldsymbol{\mathcal{Z}}_N^T]^T \in \mathbb{R}^{3N}$, $\boldsymbol{\mathcal{Z}}_e = [\boldsymbol{\mathcal{Z}}_{1e}^T, ..., \boldsymbol{\mathcal{Z}}_{Ne}^T]^T \in \mathbb{R}^{3N}$, $\tilde{\boldsymbol{\nu}} = [\tilde{\boldsymbol{\nu}}_1^T, ..., \tilde{\boldsymbol{\nu}}_N^T]^T \in \mathbb{R}^{3N}$, $\tilde{\boldsymbol{\alpha}}^* = [\tilde{\boldsymbol{\alpha}}_1^{*T}, ..., \tilde{\boldsymbol{\alpha}}_N^{*T}]^T \in \mathbb{R}^{3N}$, and $\tilde{\boldsymbol{\Lambda}} = [\tilde{\boldsymbol{\Lambda}}_1^T, ..., \tilde{\boldsymbol{\Lambda}}_N^T]^T \in \mathbb{R}^{3N}$.

Define $n = \min(n_1, n_2)$ and $\boldsymbol{\mathcal{N}}_2 = [\|\boldsymbol{\mathcal{Z}}\|, \|\boldsymbol{\mathcal{Z}}_e\|]^T \in \mathbb{R}^2$. Then, (29) is further put into

$$\begin{aligned} \dot{V}_2 \leq &-n \|\boldsymbol{\mathcal{N}}_2\|^2 + n_3 n^* \|\boldsymbol{\mathcal{N}}_2\| \|\tilde{\boldsymbol{\nu}}\| \\ &+ \|\boldsymbol{\mathcal{N}}_2\| \|\tilde{\boldsymbol{\alpha}}^*\| + \|\boldsymbol{\mathcal{N}}_2\| \|\tilde{\boldsymbol{\Lambda}}\|. \end{aligned} \qquad (30)$$

Since $\|\boldsymbol{\mathcal{N}}_2\| \geq 2(n_3 n^* \|\tilde{\boldsymbol{\nu}}\| + \|\tilde{\boldsymbol{\alpha}}^*\| + \|\tilde{\boldsymbol{\Lambda}}\|)/n$, one has $\dot{V}_2 \leq -n \|\boldsymbol{\mathcal{N}}_2\|^2/2$. It follows that the subsystem (28) is ISS. There exists a $\mathcal{KL}$ function $\mathcal{Y}_2(\cdot)$ and $\mathcal{K}_\infty$ function $\mathcal{C}^{\tilde{\boldsymbol{\nu}}}(\cdot), \mathcal{C}^{\tilde{\boldsymbol{\alpha}}^*}(\cdot)$, and $\mathcal{C}^{\tilde{\boldsymbol{\Lambda}}}(\cdot)$ satisfying $\|\boldsymbol{\mathcal{N}}_2(t)\| \leq \mathcal{Y}_2(\|\boldsymbol{\mathcal{N}}_2(0)\|, t) + \mathcal{C}^{\tilde{\boldsymbol{\nu}}}(\|\tilde{\boldsymbol{\nu}}\|) +$

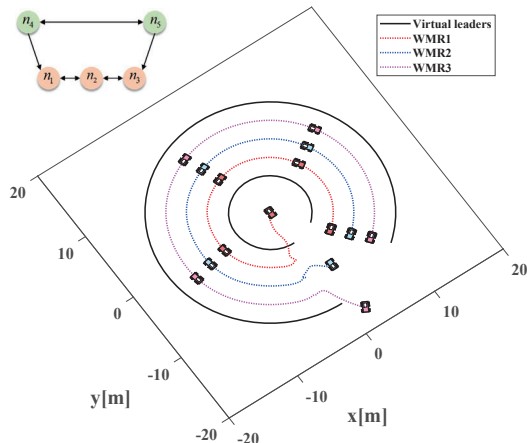

Fig. 1. Circular formation using the proposed method.

$\mathcal{C}^{\tilde{\boldsymbol{\alpha}}^*}(\|\tilde{\boldsymbol{\alpha}}^*\|) + \mathcal{C}^{\tilde{\boldsymbol{\Lambda}}}(\|\tilde{\boldsymbol{\Lambda}}\|)$, where $\mathcal{C}^{\tilde{\boldsymbol{\nu}}}(s) = 2n_3 n^* s/n, \mathcal{C}^{\tilde{\boldsymbol{\alpha}}^*}(s) = 2s/n$, and $\mathcal{C}^{\tilde{\boldsymbol{\Lambda}}}(s) = 2s/n$.

*Theorem 1:* For multi-WMRs (1) subject to initial conditions, the closed-loop system is ISS consisting of SESO (10), the DTGPG-based guidance law (22), and the SESO-based control law (26). Moreover, Zeno behavior can be avoided.

*Proof:* The ISS properties of subsystems (14) and (28) are proven through Lemma 1 and Lemma 2, respectively. The state of the subsystem (14), $\tilde{\boldsymbol{\nu}}$, and $\tilde{\boldsymbol{\Lambda}}$ are inputs of the subsystem (28). Under Assumptions 1-2, according to the cascade stability theorem, the closed-loop system is ISS. It yields that the ultimate boundedness of $\|\boldsymbol{\mathcal{N}}_2(t)\|$ as $t \to \infty$

$$\|\boldsymbol{\mathcal{N}}_2(t)\|_{t \to \infty} \leq \frac{2\|\tilde{\boldsymbol{\alpha}}^*\|}{n} + \mathcal{H}^*(\|\tilde{\boldsymbol{\nu}}_s\| \|\boldsymbol{P}_1 \boldsymbol{B}_1\| + \|\dot{\boldsymbol{\Lambda}}\| \|\boldsymbol{P}_1 \boldsymbol{C}_1\|) \quad (31)$$

with $\mathcal{H}^* = (4(n_3 n^* + 1)\sqrt{\lambda_{\max}(\boldsymbol{P}_1)})/(n j_1 \sigma_1 \sqrt{\lambda_{\min}(\boldsymbol{P}_1)})$. The detailed proof process of the Zeno behavior can be referred to [5]. The proof of Theorem 1 is complete.

## IV. SIMULATION RESULTS

From Fig. 1, it can be seen that we consider a communication topology consisting of three followers $n_1, n_2$, and $n_3$, as well as two virtual leaders $n_4$ and $n_5$ to verify the effectiveness of the proposed controller. The physical parameters of the WMR can refer to [10]. This external disturbance is similar to [16]. The initial values of three followers are chosen as $\boldsymbol{\eta}_1(0) = [0, 0, 3\pi/2]^T, \boldsymbol{\eta}_2(0) = [2, -10, \pi/2]^T, \boldsymbol{\eta}_3(0) = [2, -17, 4\pi/3]^T$. The trajectories of the two virtual leaders are chosen as

$$\begin{cases} \boldsymbol{\eta}_{4r} = [-5\sin(0.2t), -5\cos(0.2t), \mathrm{atan2}(\dot{\eta}_{4y}, \dot{\eta}_{4x})]^T \\ \boldsymbol{\eta}_{5r} = [-15\sin(0.2t), -15\cos(0.2t), \mathrm{atan2}(\dot{\eta}_{5y}, \dot{\eta}_{5x})]^T. \end{cases}$$

The main design parameters are set as $\boldsymbol{\kappa}_{11} = \mathrm{diag}\{12, 7, 10\}$, $\boldsymbol{\kappa}_{21} = \mathrm{diag}\{7, 7, 10\}, \boldsymbol{\kappa}_{31} = \mathrm{diag}\{12, 9, 10\}, \boldsymbol{\kappa}_{i2} = \mathrm{diag}\{20, 20, 20\}$, $\boldsymbol{\varepsilon}_{i1} = \mathrm{diag}\{2, 2, 2\}, \boldsymbol{\varepsilon}_{i2} = \mathrm{diag}\{40, 40, 40\}, T_{1x,a} = T_{1\psi,a} = T_{2x,a} = T_{2\psi,a} = T_{3x,a} = T_{3\psi,a} = 0.5, T_{1y,a} = T_{2y,a} =$

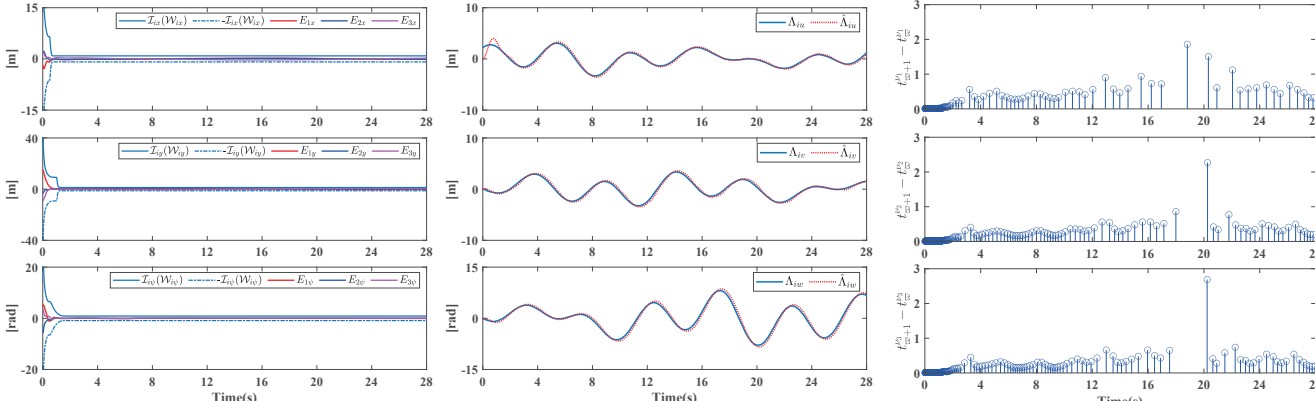

Fig. 2. Tracking errors using the DTGPG.  Fig. 3. The estimated disturbances using the SESO.  Fig. 4. The number of triggering events.

$T_{3y,a} = 1, T_{1x,b} = T_{2x,b} = T_{3x,b} = 0.7, T_{1y,b} = T_{2y,b} = T_{3y,b} = 1.2, T_{1\psi,b} = T_{2\psi,b} = T_{3\psi,b} = 1.5, \omega_{ik} = 0.7, \Theta_{ik,\infty} = 0.9, \varrho_{ik} = 2, l_{ik} = 10, \mathcal{X}_1 = \mathcal{X}_2 = \mathcal{X}_3 = 0.06.$

Simulation results are depicted in Figs 1-4. Fig. 1 demonstrates these three vehicles forming a circular formation guided by two virtual leaders. Fig. 2 shows that the tracking profile is not constrained by the initial value and is able to dynamically adjust the performance boundaries using the proposed DTGPG control scheme. Fig. 3 shows that SESO is not only able to estimate internal uncertainties and external disturbances but also to reduce chattering. Fig. 4 shows the number of triggering events. $\nu_1^\star$, $\nu_2^\star$, and $\nu_3^\star$ are triggered 179, 213, and 211 times respectively. Compared to time triggering 2800 times, it effectively saves resources.

## V. Conclusion

In this paper, the dynamic threshold global prescribed performance formation control problem was investigated for WMRs in the presence of unknown total disturbances. A dynamic threshold global performance-guaranteed formation control method based on SESO was proposed, which had three advantages: 1) it could adjust the steady-state performance boundary twice, 2) it resolved the initial value constraints present in standard PPC, and 3) it mitigated the chattering problem in event-triggered ESO. This cascade system consisting of the SESO, the DTGPG-based guidance law, and the SESO-based control law was proved to be ISS. The main results were demonstrated by the simulation examples.

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
