# OpenReview forum: "Dynamic Threshold Global Performance-Guaranteed Formation Control for Wheeled Mobile Robots with Smooth Extended State Observer"
_IEEE.org/ICIST/2024/Conference — IEEE ICIST 2024 Conference Submission_

### Official Review · Reviewer_NwDG · 2024-08-26
**In this paper, the dynamic threshold global prescribed performance formation control problem was investigated for WMRs in the presence of unknown total disturbances. A dynamic threshold global performance-guaranteed formation control method based on SESO was proposed, which had three advantages: 1) it could adjust the steady-state performance boundary twice, 2) it resolved the initial value constraints present in standard PPC, and 3) it mitigated the chattering problem in event-triggered ESO.**

**Rating:** 8
**Confidence:** 4

**Review:**

I think the conclusion section is too brief, and I believe the author has a clear understanding of the limitations and future developments of the methods proposed in the article.

---

### Official Review · Reviewer_5JM1 · 2024-08-29
**Accept after modification**

**Rating:** 7
**Confidence:** 4

**Review:**

This paper primarily proposes a dynamic threshold global performance-guaranteed formation control method for wheeled mobile robots. Overall, the manuscript provides a thorough exposition. However, a few issues warrant attention:

1.The manuscript employs two different notations for equations: "( )" and "equations ( )." It is recommended to review the entire document and standardize the notation to a single, consistent format.
2.In Equation 1, the variables R, M, F, r, and D lack corresponding nomenclature. It is suggested that the authors provide their definitions and explain their practical significance.
3.The text currently uses two variations of the term for the event triggering mechanism—"event triggering mechanism" and "event-triggered mechanism." It would be beneficial to standardize this terminology throughout the manuscript.
4.The vertical axis labels in Figures 2 and 3 should include more information than just units; please ensure that they are fully descriptive.
5.In Figure 4, the horizontal axis represents time, while the vertical axis denotes the duration of each event triggering. However, the figure is titled ‘The number of triggering events.’ This title may not accurately reflect the content of the graph. It is recommended to revise the figure title to better represent the data presented.

---

### Official Review · Reviewer_BEeo · 2024-08-30
**This paper can be accepted.**

**Rating:** 7
**Confidence:** 3

**Review:**

1. Format and alignment of author information, omit 1st and 2th.
2. The fonts of legends and labels are small. It is recommended to increase the font size and image resolution.
3. When displaying multiple data sets in the same chart, use more distinct line styles and colors.
4. The experimental settings can be described in more detail, including the settings of the simulation environment, the simulation of random interference, and the selection criteria of control parameters, etc.
5. The article mainly compares with traditional methods. Comparative experiments can be added to compare with existing technology, the latest, different backgrounds or other improvement methods to highlight the innovation and adaptability of the proposed method.
6. Add explanations of some key steps in practical applications. For example, when proving input-to-state stable (ISS), you can briefly explain the significance and impact of ISS in practical applications; in the section discussing the number of trigger events, you can further explain The significance of these triggering events in practical applications (including the relationship between resource consumption and real-time performance), etc.
7. In the final summary part, the practical application value of these innovations can be further emphasized, such as the application prospects in actual robot control, etc.

---

### Decision · Program_Chairs · 2024-09-06

Accept (Oral)